# What’s New in the Molecular Mechanisms of Diabetic Kidney Disease: Recent Advances

**DOI:** 10.3390/ijms24010570

**Published:** 2022-12-29

**Authors:** Kimio Watanabe, Emiko Sato, Eikan Mishima, Mariko Miyazaki, Tetsuhiro Tanaka

**Affiliations:** 1Dialysis Center, Tohoku University Hospital, Sendai 980-8574, Japan; 2Division of Clinical Pharmacology and Therapeutics, Faculty of Pharmaceutical Sciences, Graduate School of Pharmaceutical Sciences, Tohoku University, Sendai 980-8578, Japan; 3Division of Nephrology, Rheumatology and Endocrinology, Graduate School of Medicine, Tohoku University, Sendai 980-8575, Japan; 4Institute of Metabolism and Cell Death, Helmholtz Zentrum München, 85764 Neuherberg, Germany

**Keywords:** diabetic kidney disease, inflammation, fibrosis, metabolism, hemodynamics, SGLT2 inhibitor

## Abstract

Diabetic kidney disease (DKD) is the leading cause of chronic kidney disease, including end-stage kidney disease, and increases the risk of cardiovascular mortality. Although the treatment options for DKD, including angiotensin-converting enzyme inhibitors, angiotensin II receptor blockers, sodium-glucose cotransporter 2 inhibitors, and mineralocorticoid receptor antagonists, have advanced, their efficacy is still limited. Thus, a deeper understanding of the molecular mechanisms of DKD onset and progression is necessary for the development of new and innovative treatments for DKD. The complex pathogenesis of DKD includes various different pathways, and the mechanisms of DKD can be broadly classified into inflammatory, fibrotic, metabolic, and hemodynamic factors. Here, we summarize the recent findings in basic research, focusing on each factor and recent advances in the treatment of DKD. Collective evidence from basic and clinical research studies is helpful for understanding the definitive mechanisms of DKD and their regulatory systems. Further comprehensive exploration is warranted to advance our knowledge of the pathogenesis of DKD and establish novel treatments and preventive strategies.

## 1. Introduction

Diabetic kidney disease (DKD) is the leading cause of chronic kidney disease (CKD), including end-stage kidney disease (ESKD), and it involves tremendous medical care costs [1]. Due to the increase in the prevalence of diabetes, the prevalence of DKD is also increasing [1]. DKD develops in approximately 40% of patients with diabetes [2]. A serious concern in patients with DKD is the increased risk of all-cause mortality and cardiovascular mortality, as well as a low quality of life [1,3]. On the other hand, although the treatment options for DKD include angiotensin-converting enzyme (ACE) inhibitors, angiotensin II receptor blockers (ARBs), sodium-glucose cotransporter 2 (SGLT2) inhibitors, and mineralocorticoid receptor antagonists (MRAs), their efficacy is still limited [4]. The renal and cardioprotective effects of SGLT2 inhibitors, which are independent of glycemic control, have clinically important implications, although even with the addition of these drugs, high cardiovascular-related mortality and DKD progression remain as important problems in these patients [5,6]. A deeper understanding of the molecular mechanisms of DKD onset and progression is needed for the development of new and innovative treatments for DKD. The pathogenesis of DKD is complex and includes various pathways. The mechanisms of DKD can be broadly classified into inflammatory, fibrotic, metabolic, and hemodynamic factors [7]. In this review, we summarize the results of recent basic research, focusing on new knowledge about the molecular mechanisms of DKD, including therapeutic interventions.

## 2. Molecular Mechanisms of Diabetic Kidney Disease

### 2.1. Inflammatory Factors

Chronic inflammation plays a key role in the development of DKD. Increased expression of inflammatory cytokines, chemokines, and growth factors has been observed in renal biopsy samples from DKD patients [8]. Donath et al. stated that diabetes can be considered an inflammatory disease [9]. Various components of the immune system are changed in diabetes, including apparent changes in circulating leukocytes, which induce alterations in the levels of specific cytokines and chemokines [9,10]. In DKD, pathological changes such as mesangial expansion, thickening of the glomerular and tubular basement membranes, and podocyte loss occur due to continuation of glomerular hypertension and hyperfiltration, followed by gradual progression to glomerular sclerosis and tubular atrophy, and finally decline in renal function [11]. Hyperglycemia contributes to mesangial expansion through various cytokines and growth factors such as TGF-β, VEGF and fibronectin [12,13,14]. Since mesangial expansion is closely related to loss of renal function, the pathological significance of these cytokines and growth factors is important [14]. In this section, we focus on and summarize the recent new findings on representative inflammatory factors (TNF-α, IL-1, IL-6, IL-16, IL-18, MCP-1, and MMP-9) that are related to the pathogenesis of DKD.

#### 2.1.1. Tumor Necrosis Factor-α (TNF-α)

Multiple inflammatory cytokines affect the onset and progression of DKD [15,16]. Among them, tumor necrosis factor-α (TNF-α), which is produced by activated macrophages, plays an important role in the induction of other cytokines, chemokines, apoptosis and cytotoxic effects [7]. In 2019, Yang et al. demonstrated that TNF-α secretion was upregulated by T cell immunoglobulin domain and mucin domain-3 (Tim-3) in renal macrophages in two diabetic mouse models (streptozotocin [STZ]-induced diabetic mice and db/db mice) and confirmed that diabetic podocyte injury is associated with Tim-3-mediated activation of NF-κB/TNF-α [16]. NF-κB is an important transcription factor that is activated by several cytokines, subsequently producing inflammatory mediators such as TNF-α that contribute to the development of DKD [17]. Niewczas et al., who used data from three independent type 1 and type 2 diabetic cohorts and analyzed 194 circulating inflammatory proteins [18], found that TNF superfamily members were associated with a 10-year risk of end-stage kidney disease [18]. In 2020, Xiang et al. demonstrated that the expression of TNF-α was increased in the kidneys of diabetic nephropathy rats [15]. They also showed improvement of renal dysfunction and inhibition of inflammation (significant downregulation of TNF-α mRNA expression and reduction in the secretion of TNF-α) by the infusion of human umbilical cord-derived mesenchymal stem cells in STZ-induced diabetic nephropathy rats [15].

#### 2.1.2. Interleukin-1 (IL-1)

Within the interleukin-1 (IL-1) superfamily, IL-1β is a proinflammatory cytokine involved in the pathophysiology of DKD [19]. A possible association between hyperglycemia and infiltrating macrophages, which play an important role in the release of large amounts of IL-1β in the renal tissue in DKD, has been suggested [19]. In 2021, Han et al. demonstrated that the protein level of IL-1β increased markedly in diabetic mice, and that metformin treatment decreased the expression of IL-1β [20]. They focused on the importance of renal tubulointerstitial fibrosis in the development of DKD and studied the association between renal tubular injury and abnormal mitophagy, which is a type of autophagy that selectively eliminates disrupted and dysfunctional mitochondria [20]. They firstly reported that metformin, the adenosine monophosphate-activated protein kinase (AMPK) agonist, ameliorated renal oxidative stress and tubulointerstitial fibrosis via the AMPK-mitophagy pathway in high-fat diet and STZ-induced diabetic mice [20]. Autophagy dysfunction is thought to be involved in impairment of the AMPK-Akt pathway. Akt phosphorylation by hyperglycemia promotes autophagy dysfunction, which causes oxidative stress and results in renal inflammation in DKD [21]. Autophagy dysfunction also enhances oxidative stress through AMPK inhibition [21]. So, improving autophagy results in inhibiting oxidative stress and inflammation through AMPK activation and Akt inhibition. Recently, Dusabimana et al. found that geniposide, a natural compound that has been reported for anti-inflammatory effects, improves renal structural and functional abnormalities by reducing albuminuria, podocyte loss, glomerular and tubular injury, renal inflammation and interstitial fibrosis in diabetic nephropathy mice [22]. They also confirmed that geniposide was associated with an increase in AMPK activity to enhance autophagy and a decrease in Akt activity to block oxidative stress and inflammation [22].

#### 2.1.3. Interleukin-6 (IL-6)

Interleukin-6 (IL-6) induces neutrophil infiltration in the tubulointerstitium, which is associated with thickening of the glomerular basement membrane and podocyte hypertrophy [7]. These changes ultimately contribute to albuminuria and decreased renal function. In 2019, Li et al. reported in a diabetic nephropathy model that phosphatase and tensin homolog (PTEN), which is a negative regulator of protein kinase B and mechanistic target of rapamycin (mTOR), promotes IL-6 [23]. They found that when PTEN was modified by MEX3C-catalyzed K27-linked polyubiquitination at lysine 80, it promoted hyperglycemia-induced epithelial–mesenchymal transition, which is considered necessary for the progression of fibrosis in tubular epithelial cells [23]. More recently, in 2021, Srivastava et al. confirmed the role of IL-6 in diabetes-associated renal fibrosis using diabetic CD-1 mice [24]. They showed that IL-6 neutralization by IL-6 neutralizing antibody injection resulted in a significant reduction of fibrosis and collagen deposition in diabetic mice [24].

#### 2.1.4. Interleukin-16 (IL-16)

Interleukin-16 (IL-16) has also been implicated in the pathogenesis of DKD [7]. In 2019, An et al. measured cytokine levels in the kidney and blood circulatory system of a rhesus macaque model of DKD before and after mesenchymal stem cell treatment [25]. They found that mesenchymal stem cell treatment not only improved renal function and decreased SGLT2 expression in renal tubular cells, but also powerfully and specifically decreased the level of IL-16 [25].

#### 2.1.5. Interleukin-18 (IL-18)

Yaribeygi et al. reviewed the relationship between interleukin-18 (IL-18) and DKD and stressed that the role of IL-18 in the inflammatory process seems to be more specific than that of other cytokines [19]. The reason IL-18 is recognized as being more important than other inflammatory cytokines is that the urinary excretion and expression of IL-18 in renal tissue correlates with urinary albumin excretion and the progression of DKD [19]. In 2019, Chen et al. confirmed a significant increase in the level of IL-18 in cultured supernatants of HK-2 cells, a proximal tubular cell line derived from normal kidney that had been treated with high glucose plus palmitic acid [26]. By transcriptome analysis, they found that the disulfide-bond A oxidoreductase-like protein (DsbA-L) expression level was decreased in the kidneys of diabetic nephropathy mice [26]. Increased levels of IL-18 in HK-2 cells with high glucose plus palmitic acid levels were further increased by DsbA-L small interfering RNA, and this change was alleviated by co-treatment with an adenosine monophosphate kinase (AMPK) activator [26]. Inflammasomes play a role in initiating inflammatory cascades with respect to IL-1β and IL-18 [27]. In particular, the NLRP3 inflammasome is the most extensively studied, and it is being studied not only for kidney disease, but also for cancer and liver disorders [10].

#### 2.1.6. Monocyte Chemoattractant Protein-1 (MCP-1)

Monocyte chemoattractant protein-1 (MCP-1), also known as CC chemokine ligand 2, plays an important role in the process of inflammation, because it enhances the expression of other inflammatory factors and cells [28]. Long non-coding RNAs are defined as RNAs with >200 nucleotides that do not encode any protein. In 2019, Zhang et al. reported that silencing of a long non-coding RNA (LRNA9884) that plays an important role in the development of DKD effectively blocked MCP-1-dependent renal inflammation in db/db mice [29]. Prolyl hydroxylase domain (PHD) inhibitors are novel therapeutic drugs for renal anemia, which act by stimulating erythropoietin production through the activation of hypoxia-inducible factor (HIF) [30]. In 2020, Sugahara et al. showed that the mRNA level of MCP-1 increased by 24-fold in diabetic black and T brachyury ob/ob mice compared with wild-type mice [30]. They demonstrated through transcriptome analysis of isolated glomeruli that enarodustat, which is a PHD inhibitor, reduces the expression of MCP-1 along with reduced glomerular macrophage infiltration [30]. The same research group also reported that HIF stabilization has favorable effects in the pathogenesis of DKD through various mechanisms, such as improving renal energy metabolism and relief of oxidative stress in renal tissue [31]. Tian et al. showed that higher urinary MCP-1 is an independent predictor of CKD progression on patients with macroalbuminuric diabetic nephropathy [32]. Also, in 2021, Schrauben et al. reported that higher plasma level of MCP-1 was associated with increased risk of progression of DKD [33]. These clinical data suggest that MCP-1 is an important therapeutic target which contributes to exacerbation of DKD.

#### 2.1.7. Matrix Metalloproteinase-9 (MMP-9)

Matrix metalloproteinase-9 (MMP-9) is one of the most complex forms of matrix metalloproteinases, which are zinc dependent proteolytic metalloenzymes [34]. In 2021, Yang et al. demonstrated that the expression of MMP-9 in the proximal renal tubular epithelial cells in a STZ-diabetic rat model was significantly increased, based on previous findings that MMP-9 regulates extracellular matrix degradation during renal fibrosis [35]. More recently, in 2022, Yeh et al. confirmed that the expression of the MMP-9 protein was increased in rats with DKD induced by 5/6th nephrectomy followed by intraperitoneal administration of aminoguanidine and STZ [36]. This increase in MMP-9 was relieved by treatment with intrarenal arterial transfusion of human umbilical cord-derived mesenchymal stem cells, resulting in improvement of creatinine and urinary protein levels [36].

### 2.2. Fibrotic Factors

Fibrosis is a phenomenon characterized by the accumulation of myofibroblasts, which are collagen-depositing cells [37]. In addition to bone marrow-derived myofibroblasts, resident fibroblasts play an important role in the development of renal fibrosis [37]. In particular, tubulointerstitial fibrosis is the final common pathway for all renal diseases, including DKD, and is associated with tubular atrophy and extracellular matrix accumulation [38]. This section focuses on and summarizes the new findings on representative fibrotic factors in relation to DKD, such as transforming growth factor-β (TGF-β), fibronectin, collagen-1, and connective tissue growth factor (CTGF).

#### 2.2.1. Transforming Growth Factor-β (TGF-β)

The central role of TGF-β in the initiation and progression of kidney fibrosis is well understood. Wang et al. summarized the role of TGF-β, which is considered a master regulator of DKD, especially in terms of inflammation and fibrosis [39]. They concluded that TGF-β and its downstream Smad molecules form a key pathway in the pathogenesis of DKD, with TGF-β1 and Smad3 being particularly pathogenic [39]. In 2021, Yang et al. showed that Smad3 promotes autophagy dysregulation by triggering lysosome depletion in tubular epithelial cells under diabetic conditions [40]. Recently, Hong et al. focused on the role of TGF-β-induced angiogenesis in DKD and CKD [41,42]. They demonstrated that leucine-rich α-2-glycoprotein 1 (LRG1), which is one of the top upregulated genes that exerts proangiogenic effects through enhancement of TGF-β1 signaling, is localized predominantly in glomerular endothelial cells, and that its expression is elevated in the kidneys of unilaterally nephrectomized, STZ-induced DKD mice [41].

#### 2.2.2. Fibronectin

Accumulation of fibronectin, which is ubiquitously expressed in the extracellular matrix, in the glomerular mesangium in DKD is associated with deterioration of kidney function [12]. In 2017, Klemis et al. showed that circulating fibronectin contributes to mesangial expansion and exacerbation of albuminuria in STZ-induced DKD mice [12]. Using liver-specific conditional-knockout mice, they also confirmed that a 90% decrease in circulating fibronectin resulted in a decrease in mesangial expansion and a decline in albuminuria [12]. In 2021, Lin et al. performed a systematic review and meta-analysis to evaluate the therapeutic effects of mesenchymal stem cell therapy in DKD [43]. Treatment of DKD with mesenchymal stem cell therapy significantly decreased fibrosis indicators, including fibronectin, and improved renal function (blood urea nitrogen and serum creatinine) and urinary albumin [43].

#### 2.2.3. Collagen-1

In 2021, Han et al. confirmed that in high-fat diet and STZ-induced DKD mice, collagen-1 is enhanced in both renal immunostaining and protein expression [20]. Furthermore, they reported that metformin significantly decreased collagen-1 along with fibronectin [20]. Yang et al. used immunostaining to confirm the enhancement of collagen-1 and real time-PCR to confirm the increase in mRNA expression in mice with high-fat diet and STZ-induced type 2 diabetes [44]. Furthermore, they reported that knockdown of AT-rich interactive domain 2-IR (Arid2-IR), a Smad3-related long non-coding RNA (lncRNA), improved the high expression level of collagen-1 in DKD mice [44]. These studies suggest that collagen-1 is deeply involved in the pathogenesis of renal fibrosis and excessive accumulation of extracellular matrix in DKD.

#### 2.2.4. Connective Tissue Growth Factor (CTGF)

The role of connective tissue growth factor (CTGF) in fibrosis has been confirmed in a human renal biopsy study [45]. The study found that CTGF expression levels correlated with the degree of glomerulosclerosis and tubulointerstitial fibrosis in patients with various renal diseases [45]. Furthermore, it was confirmed that urinary CTGF concentrations in diabetic patients correlated with higher urinary albumin excretion and lower eGFR [45]. As mentioned in Section 2.1.3, Li et al. confirmed in 2019 that PTEN enhances not only IL-6, but also CTGF in DKD model mice [23]. In 2022, Looker et al. reported in an observational cohort study that higher serum PTEN (particularly PTEN^K27polyUb^) is associated with a risk of decreased GFR in patients with DKD [46]. DsbA-L, which has been identified in the matrix of rat liver mitochondria, is suggested to play a protective role in lipid-related kidney injury in DKD [26]. In 2020, Chen et al. focused on yes-associated protein (YAP), a transcriptional coactivator for numerous target genes in the nucleus, and found that it is involved in CTGF production and release [47]. This study demonstrated that YAP plays an important role in diabetic interstitial fibrogenesis [47].

### 2.3. Metabolic Factors

#### 2.3.1. Reactive Oxygen Species (ROS)

Hyperglycemia promotes the production of reactive oxygen species (ROS), and excess ROS in podocytes has a profound effect on the progression of DKD [48,49]. Evidence so far suggests that ROS are produced by various inflammasomes, including nucleotide leukin-rich polypeptide 3 (NLRP3), and also contribute to the formation and activation of inflammasomes [50]. In addition, ROS promote DKD by inducing epithelial-to-mesenchymal transition (EMT) and apoptosis [51,52]. In 2021, Wu et al. reported that knocking down of NLRP3 prevented hyperglycemia-induced mitochondrial ROS in podocytes [53]. These data indicate that inhibition of NLRP3 inflammasome activation prevents podocyte injury in DKD by reducing mitochondrial ROS generation. In 2021, Hou et al. reported that CD36 promotes NLRP3 inflammasome activation in diabetic db/db mice via the mitochondrial ROS pathway in renal tubular epithelial cells [54]. CD36, which is also known as scavenger receptor B2, is a multifunctional receptor that plays roles in inflammatory signaling, apoptosis, and kidney fibrosis [55]. CD36 is expressed in tubular epithelial cells, podocytes, and mesangial cells, and is significantly upregulated in CKD [55]. Hou et al. also confirmed in an in vivo experiment that inhibition of CD36 could protect diabetic mice from tubulointerstitial inflammation [54]. In 2020, Shi et al. showed hyperglycemia-induced elevation of ROS and matrix protein accumulation in renal mesangial cells [56]. They further suggested that human antigen R (HuR), a ubiquitously expressed RNA-binding protein that is partially involved in EMT in DKD, is involved in ROS elevation and matrix protein accumulation in renal mesangial cells [56]. In 2022, Jha et al. demonstrated the pathological role of the pro-oxidant enzyme NADPH oxidase 5 (NOX5) [57]. They found that overexpression of NOX5 in DKD mice enhances albuminuria, renal fibrosis, and inflammation via increased ROS formation [57].

#### 2.3.2. Advanced Glycation End Products (AGEs)

Advanced glycation end products (AGEs) contribute to tissue damage in DKD by two major mechanisms: alteration of extracellular matrix architecture, and modulation of cellular functions through interaction with the receptor for AGEs (RAGE) [58]. N (epsilon)-(carboxymethyl)-lysine (CML) is a major AGE in renal basement membranes, and its accumulation involves upregulation of RAGE on podocytes in DKD [58]. Accumulating evidence since the identification of RAGE indicates that AGEs/RAGE play a central role in the pathogenesis of DKD [59,60]. Sustained hyperglycemia induces AGE formation, and the interaction between AGEs and RAGE induces oxidative stress [59]. In fact, several reports have shown that inhibition of AGEs or AGE/RAGE signaling improves the pathology of DKD in animal models. Hou et al. reported that Salvianolic acid A, which is a water-soluble phenolic acid extracted from the dried root and rhizome of *Salvia miltiorrhiza* Bunge, inhibits the oxidative stress induced by AGEs [61]. They demonstrated that Salvianolic acid A significantly reduced urinary albumin and ameliorated renal dysfunction via reduction of endothelial loss, glomerular hyperfiltration, glomerular hypertrophy, and mesangial matrix expansion [61]. Watson et al. treated diabetic apolipoprotein E knockout mice with an AGE inhibitor (alagebrium), and confirmed that alagebrium reduces glomerular fibrogenesis and inflammation beyond preventing RAGE activation [62]. Matsui et al. screened DNA aptamers directed against RAGE in vitro, and examined the effects on the development and progression of DKD in STZ-induced diabetic rats [63]. They confirmed that continuous intraperitoneal administration of RAGE-aptamer to DKD rats improved podocyte damage and significantly decreased albuminuria [63]. Recently, in 2021, Azegami et al. developed a vaccine against RAGE and showed that RAGE vaccination significantly decreased urinary albumin excretion in type 1 and type 2 diabetic mouse models [64]. They also confirmed that RAGE vaccination suppresses glomerular hypertrophy, mesangial expansion, and glomerular basement membrane thickness [64]. As mentioned above, AGE accumulation and its interaction with RAGE play an important role in the pathogenesis of DKD, and its inhibition is thought to be an attractive treatment. However, the results of clinical studies focusing on RAGE antagonism or AGE inhibitors have remained controversial for decades [65,66,67,68]. In the future, RCTs with large sample sizes are expected to clarify their efficacy for the development and progression of DKD.

#### 2.3.3. Gut Microbiome Changes

Exposure to AGEs is partly derived from diet, as well as from hyperglycemia. AGEs are induced by hyperglycemia and oxidative stress [69]. AGEs directly and indirectly (through their receptor, RAGE) cause further oxidative stress, NF-κB mediated inflammation, and fibrosis via TGF-β [69]. Vlassara, H et al. reported that AGEs caused pathological changes of DKD in rats, such as glomerular hypertrophy, mesangial expansion, glomerular basement membrane thickening, and glomerular sclerosis [70]. Therefore, the role of hyperglycemia-induced AGEs in DKD pathogenesis is important. In DKD patients with decreased GFR, the pathological significance of AGEs becomes more important, because their clearance and excretion are decreased. Although, the clinical impact of AGE-rich foods remains uncertain, various experimental attempts have been made in terms of gut microbiome changes and inflammation. Dietary AGEs interact with colonic microbiota and trigger local inflammation and the release of inflammatory mediators [71]. It has been suggested that AGE-rich foods, such as processed foods, disrupt intestinal barrier permeability and translocation of proinflammatory mediators into the systemic circulation, resulting in local kidney inflammation [72]. Lipopolysaccharides from the cell wall of Gram-negative bacteria in the gut bind to toll-like receptor (TLR) -4 and contribute to increased local cytokine production and recruitment of inflammatory cells [73]. TLR-4 has been reported to promote inflammation in DKD [74], and progression of DKD was suppressed in TLR-4-deficient STZ diabetic mice [74]. Specifically, urinary albumin was significantly decreased, and glomerular hypertrophy and hypercellularity, as well as podocyte and tubular injury were suppressed [74]. In 2019, Kikuchi et al. focused on phenyl sulfate, which is a gut microbiota-derived metabolite, and showed that it contributes to albuminuria and podocyte damage in DKD rats [75]. More recently, in 2022, Linh et al. showed that impaired mitochondrial antiviral signaling protein (MAVS), which is a component of innate immunity and maintains intestinal integrity, contributes to DKD progression via intestinal barrier dysfunction [76]. As mentioned above, disturbance of the normal gut microbiota has been implicated in the pathogenesis of DKD. In 2020, using STZ-induced diabetic mice, Li et al. showed that dietary fiber protects against DKD [77]. They focused on the reduction of short-chain fatty acids (SCFAs) to evaluate dysbiotic changes in the gut in DKD [77]. They confirmed that albuminuria, glomerular hypertrophy, podocyte injury, and interstitial fibrosis were significantly improved by a high-fiber diet compared to control mice through the improvement of gut microbiota, which promoted the growth of SCFA-producing bacteria and increased fecal and systemic SCFA concentrations [77].

### 2.4. Hemodynamic Factors

Glomerular hyperfiltration is a characteristic pathology in the early stages of DKD, and glomerular enlargement is a characteristic histological finding in DKD and obesity-related nephropathy [1]. In classical and typical diabetic nephropathy, glomerular hyperfiltration is the first stage in the pathogenesis, leading to progressive albuminuria, declining GFR, and finally end-stage kidney disease (ESKD), although the mechanism is not fully understood [78,79,80]. Glomerular hypertension is also influenced by lifestyle factors, such as diet and body weight. Hyperglycemia interacts with elevated levels of circulating amino acids caused by a high-protein diet and becomes a trigger for glomerular hyperfiltration [81]. SGLT2 is a postulated mechanism of glomerular hyperfiltration. SGLT2 increases the reabsorption of glucose in the proximal tubules, thereby reducing the delivery of sodium chloride to the macula densa [81,82,83,84,85]. As a result, tubulo-glomerular feedback is reduced, afferent arterioles are dilated, and angiotensin II is increased in efferent arterioles, resulting in vasoconstriction [81,82]. These effects increase glomerular perfusion and intraglomerular pressure, leading to glomerular hyperfiltration. On the other hand, it has been suggested that the renal and cardioprotective effects of SGLT2 inhibitors are due to factors beyond the correction of hemodynamics and blood glucose levels [86,87,88,89]. In this section, we focus on the renin–angiotensin system (angiotensin II, aldosterone) and endothelin as hemodynamic factors that contribute to the development of DKD.

#### 2.4.1. Renin–Angiotensin–Aldosterone System (RAAS)

Along with SGLT2, the renin–angiotensin–aldosterone system (RAAS) is an important upstream mechanism involved in the progression of DKD [79,90]. Angiotensin II is one of the bioactive members of the RAAS and has been confirmed to be elevated in DKD [91]. Angiotensin II has various physiologic effects and is known to promote ROS production [92]. In 2018, Ilatovskaya et al. reported that podocyte damage via angiotensin II-mediated calcium influx into podocytes contributes to renal injury in DKD [93]. Kidney enlargement is often seen in DKD. The mechanism of kidney enlargement is related to elongation of the proximal tubules caused by an increase in single nephron GFR and glomerular hypertension [79]. Although it was suggested that RAAS activation is deeply involved in the pathogenesis of DKD in terms of the glomerular component, how the presence of RAAS in the renal tubules affects the pathogenesis of DKD remains unclear. In 2022, using an STZ-induced DKD mouse model, Haruhara et al. found that RAAS hyperactivation in kidney tubules exacerbates diabetic glomerular injury [94]. They also showed that tubulointerstitial macrophage polarization attenuates TNF-alpha and oxidative stress in the progression of DKD, and moderates the tubulo-glomerular interaction [94]. Aldosterone is also plays an important roles in the pathophysiology of DKD [95]. Ritz et al. reported that aldosterone upregulates unfavorable growth factors such as plasminogen activator inhibitor 1 and TGF-β, which promote macrophage infiltration and consequent renal fibrosis [96].

#### 2.4.2. Endothelin (ET)

Endothelin (ET) is a potent renal vasoconstrictor, modulating renal blood flow and glomerular filtration [97]. ET has three isoforms, ET-1, ET-2, and ET-3, of which ET-1 has major biological activity [97]. ET-1 has been shown to activate proinflammatory and profibrotic pathways, and its plasma level increases in response to hyperglycemia, endothelial dysfunction, and oxidative stress [7,98]. In addition, increased renal ET expression is associated with mesangial proliferation and podocyte injury [99]. Through these mechanisms, ET is involved in the pathogenesis of DKD, as has been confirmed in various experimental animal models [100]. Renal protective effects, including reduction of albuminuria and improvement of renal morphology, as well as the roles of ET-1 inhibition and ET-1 receptor antagonism in experimental models of DKD, have also been reported [100,101].

As described above, the molecular mechanisms involved in the onset and progression of DKD is summarized (Figure 1). Also, mechanisms of action in DKD onset and progression and characteristic findings is summarized (Table 1).

### 2.5. Recent Advances in the Treatment of DKD

Based on the molecular mechanisms of DKD, several therapeutic drugs have been developed for clinical use. In this section, we mainly outline the drugs whose clinical effects have actually been confirmed as therapeutic agents for DKD and for which new findings have been reported in recent years.

#### 2.5.1. SGLT-2 Inhibitors

Various mechanisms have been postulated for the effects of sodium-glucose cotransporter-2 (SGLT2) inhibitors on DKD, including activation of tubulo-glomerular feedback, decrease in the circulating levels of IL-6, TNF receptor-1, matrix metalloproteinase-7, and fibronectin-1, and reduced ketone production [84,86,102,103,104,105,106]. Anti-inflammatory and antifibrotic effects were suspected to be comprehensively induced by these factors. Worldwide, clinical practice guidelines recommend treatment of most DKD patients with SGLT2 inhibitors regardless of the degree of glycemic control, based on the results of major randomized-control trials (RCTs) and their systematic review and meta-analysis [107,108]. Among them, the results of two large RCTs, the CREDENCE and DAPA-CKD trials, have had a strong impact on the clinical utility of SGLT-2 inhibitors [109,110]. The CREDENCE trial showed a 34% reduction in risk of renal composite outcomes, which include end-stage kidney disease, a doubling of the creatinine level, or death from renal causes in patients with diabetes and CKD [109]. Additionally, the DAPA-CKD trial showed a 44% reduction in risk of renal composite outcomes, which include a sustained decline in the estimated GFR of at least 50%, end-stage kidney disease, or death from renal causes in patients with CKD, regardless of the presence or absence of diabetes [110]. SGLT2 inhibitors can reduce the risk of important kidney endpoints, including ESKD. Since the absolute reduction in risk of kidney events was greater in patients with increased albuminuria (such patients have a higher risk of developing major kidney events), the recommendation is stronger in DKD patients with severe albuminuria [108,111]. Thus, SGLT-2 inhibitors have been identified as central and important drugs for improving renal and cardiovascular event risks in patients with DKD. Furthermore, in 2022, a large scale multicenter international RCT (the EMPA-KIDNEY trial) revealed the renal protective effects of SGLT2 inhibitors even in non-DKD CKD patients [112], with their efficacy being even more prominent in those in whom the albumin-to-creatinine ratio was more than 300 mg/g [112].

#### 2.5.2. Nonsteroidal Mineralocorticoid Receptor Antagonists (MRAs)

Mineralocorticoid receptor activation induces the activation of inflammatory and fibrotic factors in podocytes and mesangial cells [4]. Specifically, increased expression of inflammatory factors, such as IL-6, IL-1β, IL-18, TNF-α, and MCP-1, and activation of inflammatory pathways through NF-κB have been reported [113]. Pharmacological inhibition of mineralocorticoid receptors was shown to reduce albuminuria, kidney inflammation, and fibrosis in basic research using DKD model animals [4]. The 2022 guidelines of the American Diabetes Association (ADA) and the Kidney Disease Improving Global Outcomes (KDIGO) for the treatment of patients with DKD advise the use of finerenone, which is a nonsteroidal mineralocorticoid receptor antagonist (MRA), in patients who have increased albuminuria despite treatment with an ACEi/ARB and SGLT2 inhibitor [114,115]. The reasons for recommending a nonsteroidal MRA in both of these clinical practice guidelines are the results of a recent RCT (FIDELIO-DKD clinical trial) [116], which evaluated renal outcomes (reduction of eGFR >40% or death due to renal disease) in 5734 DKD patients during a median follow-up of 2.6 years [116], and demonstrated an 18% reduction in risk of renal prognosis with finerenone as the primary outcome [116]. Furthermore, the FIDELITY pooled analysis by the FIDELIO-DKD and FIGARO-DKD investigators showed a 23% reduction in risk of composite kidney outcomes, which include a sustained >57% decrease in eGFR from baseline over 4 weeks or renal death [117]. Owing to this analysis, the clinical efficacy of finerenone was confirmed across a broad spectrum of DKD stages. In 2021, Kolkhof et al. confirmed that combination therapy of finerenone and empagliflozin in hypertensive rats resulted in reduction of albuminuria and kidney fibrosis [118]. However, clinical data such as post hoc analysis of DAPA-HF [119] and the EMPEROR-reduced trial [120] did not show the efficacy of combination of these two drugs.

#### 2.5.3. Glucagon-like Peptide-1 Receptor (GLP-1R) Agonists

Glucagon-like peptide-1 receptor (GLP-1R) agonists have been suggested to have renoprotective effects, and the mechanisms involved include attenuation of oxidative stress, fibrosis, and cellular apoptosis [121]. GLP-1R agonists reduce the production of ROS and inhibit the binding of NF-κB to its target genes, which reduces the downstream expression of cytokines (TNF-α, IL-1β, IL-6 and TGF-β) and fibrotic factors [121]. In 2021, a systematic review and network meta-analysis of RCTs by Palmer et al. showed that GLP-1R agonists reduced the risk of eGFR <15 mL/min/1.73 m^2^ and the need to start kidney replacement treatment by 22% within 5 years [108]. Additionally, in 2021, Gertstein et al. showed the renoprotective effects (in terms of decrease in kidney function or development of macroalbuminuria) of GLP-1R in a large-scale international RCT [122]. In that study, which included 4076 participants with at least one risk factor for CVD, including CKD, the GLP-1R agonist efpeglenatide reduced the risk of composite renal outcomes by 32% [122]. More recently, in 2022, Wright et al. demonstrated that a combination treatment regimen of SGLT2 inhibitor and GLP-1R agonists was associated with a lower risk of major adverse cardiac and cerebrovascular events and heart failure in patients with type 2 diabetes compared to each drug alone in three nested case–control studies conducted in England and Wales [123]. In the future, RCTs are expected to verify whether the combination of these two drugs is effective for the onset and progression of DKD.

#### 2.5.4. Endothelin Receptor Antagonists (ERAs)

Endothelin (ET) has three isoforms, ET-1, ET-2, and ET-3 [97]. Among them, ET-1 plays a central role in activating inflammatory factors and fibrotic factors and promoting DKD [98]. Atrasentan, a selective ET-A receptor antagonist (ERA), has been shown to reduce albuminuria in basic and clinical studies [100,101]. As for the mechanism, it has been shown that atrasentan improves endothelial function by restoration of endothelial glycocalyx, which contributes to the decrease in albuminuria [100]. The most recent RCT that evaluated the renoprotective effect of ERAs in DKD patients was the SONAR trial, which was published in 2019 [124]. The study randomized 2648 patients with DKD 1:1 to the atrasentan or placebo group with a median follow-up of 2.2 years, and assessed doubling of serum creatinine and end-stage kidney disease as the primary composite renal endpoint events [124]. As a result, atrasentan reduced the primary composite renal endpoint event rate by 35% compared with placebo, but increased the rate of hospitalization for heart failure due to fluid retention by 33% [124]. The authors concluded that although ERAs might be effective in DKD patients with a low risk of heart failure, it is unclear whether their effectiveness outweighs the risks of heart failure and anemia. Currently, the KDIGO 2022 Guideline for Diabetes Management in CKD makes no specific recommendations on the use of ERAs [115].

#### 2.5.5. Other Agents

Bardoxolone methyl exhibits antioxidative and anti-inflammatory effects, which are based on activation of the Keap1-Nrf2 pathway [125]. In a phase 2 RCT (TSUBAKI study) conducted in Japan, bardoxolone methyl significantly increased GFR during the 16 week treatment period only in the treatment group [126]. A new large-scale phase 3 RCT (AYAME study) for evaluating the long-term efficacy and safety of bardoxolone methyl is currently being conducted in Japan [127]. The study is now ongoing and is registered at ClinicalTrials.gov (NCT03550443).

A placebo-controlled phase 2 trial in moderate-to-advanced DKD patients suggested that selonsertib, which is an apoptosis signal-regulating kinase 1 (ASK1) inhibitor, might suppress the progression of DKD [128]. A larger RCT is currently being conducted to examine the efficacy and safety of selonsertib. This ongoing study is registered at ClinicalTrials.gov (NCT04026165) [129].

### 2.6. Significance of Molecular Mechanisms as Biomarkers and Therapeutic Targets

As described above, the molecular mechanisms involved in the onset and progression of DKD are complex (Figure 2). It is important to consider how each molecule and pathway affect DKD, especially in humans. In other words, we need to consider the significance of molecular mechanisms both as biomarkers and therapeutic targets. We summarized factors involved in DKD pathogenesis in Table 1. Among them, RAAS has significance as a therapeutic target rather than a biomarker. The efficacy of angiotensin II receptor blocker, angiotensin converting enzyme inhibitor and non-steroidal mineralocorticoid receptor antagonist (finerenone) for DKD has been widely confirmed in various large clinical trials so far [130]. Endothelin also has significance as a therapeutic target rather than a biomarker, as described Section 2.5.4. Endothelin Receptor Antagonists (ERAs). AGEs are also considered therapeutic targets; however, the results of clinical studies focusing on RAGE antagonism or AGE inhibitors have remained controversial for decades [66,67,68,69]. On the other hand, plasma levels of TNF, MMP, IL-6 and fibronectin were used as inflammatory or fibrotic biomarkers in a clinical trial [103]. Although other factors also have been shown to be involved in the mechanisms of DKD, their significance as therapeutic targets or biomarkers remains unclear.

## 3. Conclusions

We summarized the recent findings of basic research focusing on new findings and developments regarding the molecular mechanisms of DKD, including therapeutic interventions. New findings and therapeutic methods for DKD are continuously being discovered in each of the fields of inflammation, fibrosis, metabolism, and hemodynamics. Collective evidence from basic and clinical research studies is helpful for understanding of the definitive mechanisms of DKD and their regulatory systems. Further comprehensive exploration is needed to advance our knowledge of the pathogenesis of DKD and to establish novel treatment and preventive strategies.

## Figures and Tables

**Figure 1 ijms-24-00570-f001:**
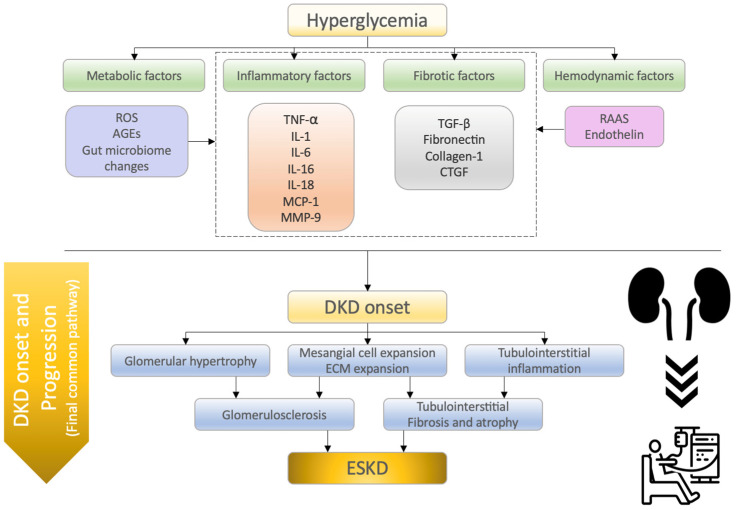
Summary of the molecular mechanisms of diabetic kidney disease (DKD) onset and progression. Several mechanisms contribute to the pathogenesis of DKD, including metabolic, inflammatory, fibrotic, and hemodynamic factors. These factors comprehensively induce albuminuria and reduce renal function. Finally, DKD patients develop ESKD through a final common pathway (glomerulosclerosis, tubulointerstitial fibrosis, and atrophy). ROS, reactive oxygen species; AGEs, advanced glycation end products; TNF-α, tumor necrosis factor-α; IL-1, interleukin-1; IL-6, interleukin-6; IL-16, interleukin-16; IL-18, interleukin-18; MCP-1, monocyte chemoattractant protein-1; MMP-9, matrix metalloproteinase-9; TGF-β, transforming growth factor-β; CTGF, connective tissue growth factor; RAAS, renin–angiotensin–aldosterone system; DKD, diabetic kidney disease; ESKD, end-stage kidney disease.

**Figure 2 ijms-24-00570-f002:**
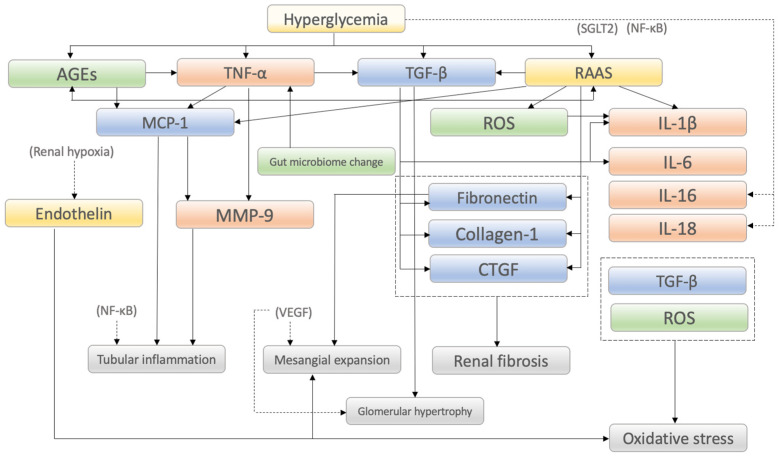
Summary of the molecular interplay that affects onset and progression of diabetic kidney disease (DKD). SGLT2, sodium glucose cotransporter 2; NF-κB, nuclear factor-kappa B; AGEs, advanced glycation end products; TNF-α, tumor necrosis factor-α; TGF-β, transforming growth factor-β; RAAS, renin–angiotensin–aldosterone system; MCP-1, monocyte chemoattractant protein-1; ROS, reactive oxygen species; IL-1β, interleukin-1β; IL-6, interleukin-6; IL-16, interleukin-16; IL-18, interleukin-18; MMP-9, matrix metalloproteinase-9; CTGF, connective tissue growth factor.

**Table 1 ijms-24-00570-t001:** Summary of factors involved in the onset and progression of diabetic kidney disease (DKD).

Category/ Factors	Mechanisms of Action in DKD Onset and Progression and/or Characteristic Findings	Reference
Inflammatory factors		
TNF-α	Produced by activated macrophages and induces other cytokines, chemokines, apoptosis and cytotoxic effects	[7,15,16,17,18]
IL-1	IL-1β plays a significant role in the association between hyperglycemia and macrophage infiltration	[19,20]
IL-6	Induces neutrophil infiltration of the tubulointerstitium and is associated with thickening of the GBM and podocyte hypertrophy	[7,23,24]
IL-16	Immunomodulatory cytokine that correlates with the severity of DKD, although concise mechanisms have not been clarified	[7,25]
IL-18	Correlates most strongly with DKD severity among cytokines and is induced by inflammasome (NLRP3)	[19,26,27]
MCP-1	Enhances the expression of other inflammatory factors such as inflammatory cytokines and inflammatory cells such as monocytes/macrophages	[28,29,30]
MMP-9	Regulates extracellular matrix degradation during fibrosis in the proximal renal tubular epithelial cells	[34,35,36]
Fibrotic factors		
TGF-β	Master regulator of inflammation and fibrosis. TGF-β1 and Smad3 (downstream of TGF-β) are particularly pathogenic	[39,40,41,42]
Fibronectin	Accumulation in glomerular mesangial lesions is associated with deterioration of kidney function	[12,43]
Collagen-1	Deeply involved in the pathogenesis of renal fibrosis and excessive accumulation of extracellular matrix	[20,44]
CTGF	Its expression level correlates with the degree of glomerulosclerosis and tubulointerstitial fibrosis	[23,45,46]
Metabolic factors		
ROS	Excess in podocytes promotes DKD. Induces epithelial-to-mesenchymal transition and apoptosis	[48,49,50,51,52,53,54,55,56,57]
AGEs	Alters extracellular matrix architecture. Modulation of cellular functions through interaction with RAGE	[58,59,60,61,62,63,64]
Gut microbiome changes	Dietary AGEs interact with colonic microbiota and trigger local inflammation and release of inflammatory mediators	[71,72,73,74,75,76,77]
Hemodynamic factors		
RAAS	Induces ROS production and podocyte damage via angiotensin-II mediated calcium influx into podocytes	[81,91,92,93,94]
Endothelin	ET-1 activates proinflammatory and profibrotic pathways and induces endothelial dysfunction and oxidative stress	[7,97,98,99,100,101]

DKD, diabetic kidney disease; TNF-α, tumor necrosis factor-α; IL-1, interleukin-1; IL-6, interleukin-6; IL-16, interleukin-16; IL-18, interleukin-18; MCP-1, monocyte chemoattractant protein-1; MMP-9, matrix metalloproteinase-9; TGF-β, transforming growth factor-β; CTGF, connective tissue growth factor; RAAS, renin–angiotensin–aldosterone system; ROS, reactive oxygen species; AGEs, advanced glycation end products; GBM, glomerular basement membrane; NLRP3, NOD-, LRR-, and pyrin domain-containing protein 3; RAGE, receptor for AGEs; ET-1, endothelin-1.

## Data Availability

Not applicable.

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
