# Peer review of "What’s New in the Molecular Mechanisms of Diabetic Kidney Disease: Recent Advances"

_ijms, 2022, doi:10.3390/ijms24010570_

Round 1

Reviewer 1 Report

Kimio Watanabe et al made an attempt to compile the molecular mechanisms of DKD onset and progression which is necessary for the development of new and innovative treatments for DKD. Manuscript is well written. However few questions can be raised.

1.       The authors cite reference 14 (Section 2.1.1) in which one of the sentences, ‘All these proteins had a systemic, non-kidney source.’ follow immediately the statement by the authors about ‘TNF superfamily members were associated with a 10-year risk of end-stage kidney disease’.  The manuscript also lists several systemic sources of mediators involved in DKD. While several mediators such as serum urea, creatinine and others involve the kidneys that are routinely analyzed during clinical evaluation of many systemic disorders, why do the authors suppose such a feature is currently out of reach in evaluating DKD?

2.       The authors mention AMPK mediation in amelioration of renal oxidative stress in Section 2.1.2 and in the following section about the role of PTEN, an Akt antagonist in promoting IL6. While it is conventionally known Akt and AMPK exert differential effects, how would such a scenario play out in DKD with an inflammatory focus? It would be helpful for a detailed account on this interplay relevant to kidney disease.

3.       The authors bring out yet another interesting yet apparently opposing role of angiogenesis in DKD pathophysiology, albeit without further discussion. In section 2.1.6, the authors put forth stabilization of HIF1α alleviates renal oxidative stress, and in Section 2.2.1 briefly discuss ‘TGF-β-induced angiogenesis in DKD and CKD…… that leucine-rich α-2-glycoprotein 1…..exerts proangiogenic effects through enhancement of TGF-β1 signaling, is localized predominantly in glomerular endothelial cells, and that its expression is elevated in the kidneys of unilaterally nephrectomized, STZ-induced DKD mice’. How do the authors view the supposedly conflicting role of angiogenesis especially during inflammation affects DKD?

4.       It would be really helpful if the authors provide a brief explanation with supporting references in the pathophysiological role of mesangial expansion in the context of DKD.

5.       Section 2.3.3 reads, ‘Exposure to AGEs is partly derived from diet, as well as from hyperglycemia’. While the authors describe the role with relevant examples of dietary or gut microbiota derived metabolites that regulate DKD. It would benefit the readers if the authors would draw the distinction and elaborate further on hyperglycemia induced AGEs in DKD pathogenesis.

6.       The authors describe the different currently available drug therapies in DKD treatment that majorly target the oxidative stress and hemodynamic factors covered in the first sections of the manuscript. Are drugs that target the fibrotic and metabolic (such as gut microbiome metabolites) available for treating DKD or related kidney diseases?

7.       The title of the manuscript reads, ‘What’s new in the molecular mechanisms of diabetic kidney disease: recent advances’. Do the authors envisage the interplay in the molecular mechanisms of different factors outlined in the manuscript that possibly can change the outcome to DKD? The inclusion of a schematic in this regard will also be really helpful to the readers.

Reviewer 2 Report

Watanabe et al provide a timely contemporary review of DKD.

Specific issues:

1.             It would be good to link some of the molecule’s pathways described to the human context and to potential treatments. e.g., TNF ?.  Comment on it as a biomarker and if TNF ? directed treatments could be useful in DKD.

2.             CTGF section.  Much of it relates to CKD in the absence of diabetes.  This should be clarified or deleted in a revised MS.

3.             MCP-1.  This has been targeted in DKD without positive results.  This should be addressed.

4.             2.3.2 – include RAGE in the title Discuss RAGE antagonism.  This field has been studied for decades without successful therapeutic approach.

5.             RAS:  More should be discussed about aldosterone as either within this section (2.4.1) re-named RAAS or as a separate section within the hemodynamic factors section.

6.             SGLT2i section should include the seminal CREDENCE and DAPA-CKD trials.  Ref 97 needs repair.

7.             2.5.2 section need to add FIGARO trial with renal data and the recent pooled analysis in FIDELITY.

8.             Consider conventional combinations e.g., Combining SGLT2i with GLP1 agonists.  Use of MRAs with SGLT2i.

Round 2

Reviewer 1 Report

Manuscript may be accepted in the present form

Reviewer 2 Report

Adequately revised